# Experimental Study on the Static Behavior of Reinforced Warren Circular Hollow Section (CHS) Tubular Trusses

**Wenwei Yang *** [ID]**, Jiankang Lin *, Ni-na Gao and Ruhao Yan**

College of Civil and Hydraulic Engineering, Ningxia University, Yinchuan 750021, China;
may5023753@163.com (N.-n.G.); YRHendeavor@163.com (R.Y.)
* Correspondence: nxyangww@163.com (W.Y.); linhealth@163.com (J.L.)

**Abstract:** For truss structures, the question of whether to weld hidden welds or not has been controversial. In the actual construction process of truss structures, the members are usually spot welded in place on the assembly platform, and then welded as a whole, while the hidden welds of the truss are not welded, especially for small pipe diameter trusses. Furthermore, in this study, under hidden weld unwelded conditions, two kinds of reinforcing method (adding a half outer sleeve on each joint and filling concrete into the chord members) are adopted to achieve the purpose of strengthening the truss. Therefore, this paper presents an experimental study on the static behavior of four types of Warren tubular trusses made of CHS members. These four types are (1) T-HW: The truss with hidden welds welded; (2) T-HN: The truss with hidden welds unwelded; (3) TS-AS: The truss strengthened on the basis of T-HN by adding a half outer sleeve on each joint; (4) TS-FC: The truss strengthened on the basis of T-HN by filling concrete into the top and bottom chord members. The mechanical behavior, failure mode, bearing capacity, and load-displacement of all specimens were investigated. The surface plasticity of the bottom chord member, the weld fracture around tubular joints at the bottom chord member, and the bending deformation of the bottom chord member were observed in the tests. Compared with the T-HW specimen, the load carrying capacity of the T-HN specimen decreased by 18%. On the other hand, the T-HN specimen has better deformability than the T-HW specimen. The reinforcing method of adding a half outer sleeve on each joint and filling concrete into the chord members can effectively improve the load carrying capacity and stiffness of the truss, thus reducing the overall deformation of the truss, but the reinforcing method of filling concrete into the chord members is more efficient.

**Keywords:** reinforced warren circular hollow section (CHS) tubular truss; hidden weld; static performance; finite element analysis

## 1. Introduction

Truss structures are popularly used in various engineering fields [1–4], and among them, steel tubular trusses have been widely applied in large engineering structures, such as stadiums, bridges, concert halls, and offshore platforms, because of their architecturally attractive shapes and favorable structural properties [5]. Many investigations have been carried out related to steel tubular truss structures [6–8]. In addition, a hybrid structural analysis and procedure were developed by Khademi [9,10] to enhance the load rating of railway truss bridges. The common steel tubular truss types are Warren trusses, Pratt truss, Fink truss, and Vierendeel truss, among which Warren trusses generally provide the most economical solution due to their reasonable mechanical performance and effective structural layout [11]. The joint form of Warren truss is usually K-joint and KK-joint, however,

for these two types of joints, the overlapped condition arises when one brace (overlapping brace) intersects the other brace (overlapped brace), requiring a weld between the braces. In addition, part of the overlapped brace is hidden within the overlapping brace, and the hidden toe of the overlapped brace may or may not be welded to the chord [12]. Nevertheless, the question whether hidden weld of overlapped joints (K-joint and KK-joint) welded or unwelded is not clearly mentioned in the current Chinese Standard for Design of Steel Structures (GB 50017-2017) [13]. The joints with hidden welds welded may have an increased strength, but with reduction in joint ductility and hysteretic performance [12,14–17]. It should be noted that the aforementioned studies mainly focus on single joint, while the research on the performance of overall truss is very limited on joints with and without hidden welds.

On other hand, steel tubular trusses made of circular hollow section (CHS) members usually fail due to surface plasticity or punching shear of the chord members due to low stiffness of the chord members. One of the commonly used reinforcing methods is to fill the chord members with grout or concrete. Concrete-filled steel tubular (CFST) members, as compared to their pure steel or concrete counterparts, have improved stiffness, load bearing capacity, buckling resistance [18–21], and even improved impact resistance [22,23]. In addition, structural health monitoring of CFST members [24–27] and assemblies [28,29] have attracted much attention. For CHS joints, extensive research works have been carried out on the static performance [30,31], hysteretic performance [32–34], impact performance [35,36], fatigue performance [37,38], and mechanical properties at elevated temperature [39–41], among others. However, compared with CHS joints, little investigation has been carried out on the performance of CFST joints. Packer et al. [42] analyzed the different failure modes of CFST joints and compared them to CHS joints, indicating the increased resistance attained by adding concrete in the chords of tubular joints. Sakai et al. [43], Huang et al. [44], and Feng et al. [45,46] investigated the performance of CFST joints under static loading. An experimental investigation was conducted by Yin et al. [47] on hysteretic behavior of tubular N-joints. Chen et al. [48] presented an experimental and numerical investigation on double-skin circular CHS tubular X-joints under axial compression. The failure mechanism was analyzed and new design equations were proposed. Hou et al. [49] studied the static behavior of concrete filled double skin steel tubular (CFDST) chord and CHS brace composite K-joints, and the effects of important parameters on joint strength were discussed based on experiment results. Moreover, Xu et al. [50], Qian et al. [51], and Tong et al. [52] carried out studies on the fatigue behavior of CFST joints.

The CFST structures have also been applied to arch trusses in bridges, as well as the truss girders in steel structures [53–55]. The research on CFST truss has been widely concerned by scholars in recent years. Experimental and analytical research was carried out by Xu et al. [56] on flexural behavior of curved concrete filled steel tubular (CCFST) trusses with curved CFST chords and hollow braces. Test results indicated that the stiffness and load-carrying capacity of CCFST trusses were larger than those of CFST trusses. Zhou et al. [57] investigated the flexural behavior of 4 circular concrete-filled stainless steel tubular truss girders. The influence of the location of concrete filling on the flexural behavior of CHS stainless steel tubular truss girder was analyzed. Mujagic et al. [58] present analytical and experimental findings pertaining to the design and behavior of composite truss members with standoff screws as shear connectors. Composite beams constituted by a concrete-encased steel truss welded to a continuous steel plate were analyzed using a nonlinear finite element formulation based on Newmark's classical model by Tullini et al. [59]. Huang et al. [60] tested the truss girders with different web arrangements and investigated their behaviors, drawing the conclusion that the geometry of the CHS girder joints was such that only chord face failure and punching shear failure could occur.

Although many researchers have studied the behavior of uniplanar CFST truss and CFST members, there are few investigations being carried out on the performance of Warren CFST trusses. In addition, the reinforcing method of adding a half outer sleeve on each joint was used for Warren CHS tubular trusses and contrasted with the behavior of Warren CFST trusses in this study. Subsequently, an experimental investigation was conducted in this paper on four types of Warren CHS tubular

trusses under static loading. The failure modes, load carrying capacity, overall deflection, and strain intensity of all test specimens were discussed. Finite element analysis (FEA) software was used for numerical modelling of Warren CHS tubular trusses and the results obtained by numerical analysis were compared with those of the experimental results.

## 2. Experimental Study

### 2.1. Test Specimens

A total of four types of Warren CHS tubular trusses, including the truss with hidden welding welded (T-HW), the truss with hidden welding unwelded (T-HN), the truss strengthened on the basis of T-HN by adding a half outer sleeve on each joint (TS-AS), and the truss strengthened on the basis of T-HN by filling concrete in top and bottom chord members (TS-FC), were designed according to the design guidelines given in the Chinese Standard for Design of Steel Structures (GB50017-2017) [13]. The truss configuration of all specimens is Warren truss with symmetric geometry, loading application, and boundary conditions.

The nominal dimensions of CHS members, including chord members, brace members, and lateral bracings of all types of multiplanar tubular trusses, are identical for comparison, and the overall length, width, and height are 2700 mm, 500 mm, and 450 mm, respectively. The effective span between the end supports of the bottom chord members is 2500 mm. The tubular joints are equally spaced in 500 mm increments along the truss span. The top and bottom chord members of all specimens are CHS of $\Phi 60 \times 2.5$ with an outer diameter ($D$) of 60 mm and wall thickness ($T$) of 2.5 mm. The corresponding chord diameter:chord thickness ratio ($2\gamma = D/T$) is equal to 12. The diagonal brace members of all specimens are CHS of $\Phi 33 \times 2$ with an outer diameter ($d$) of 33 mm and wall thickness ($t$) of 2 mm. The corresponding brace diameter:chord diameter ratio ($\beta = d/D$) and brace thickness:chord thickness ratio ($\tau = t/T$) are equal to 0.58 and 0.8, respectively. Besides, for every joint in all specimens, the angle between brace and chord ($\theta$) and the overlap rate between the braces ($O_v$) are equal to 60° and 39.3%, respectively. The lateral bracings of all specimens are CHS of $\Phi 22 \times 2$ with outer diameter ($d_1$) of 22 mm and wall thickness ($t_1$) of 2 mm. The welds connecting brace and chord members, lateral bracing, and chord members were designed according to the Chinese national standard for the welding of a steel structure (JGJ81-2002) [61], using advanced welders with full penetration, and the weld legs were twice the wall thickness of the diagonal brace. The dimensions of all specimens including chord members, brace members, and lateral bracings are detailed in Figure 1 and listed in Table 1.

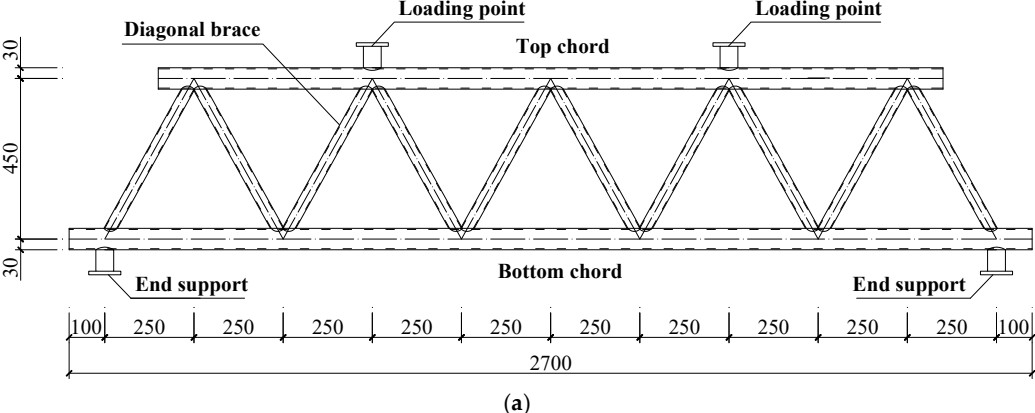

(a)

**Figure 1.** *Cont.*

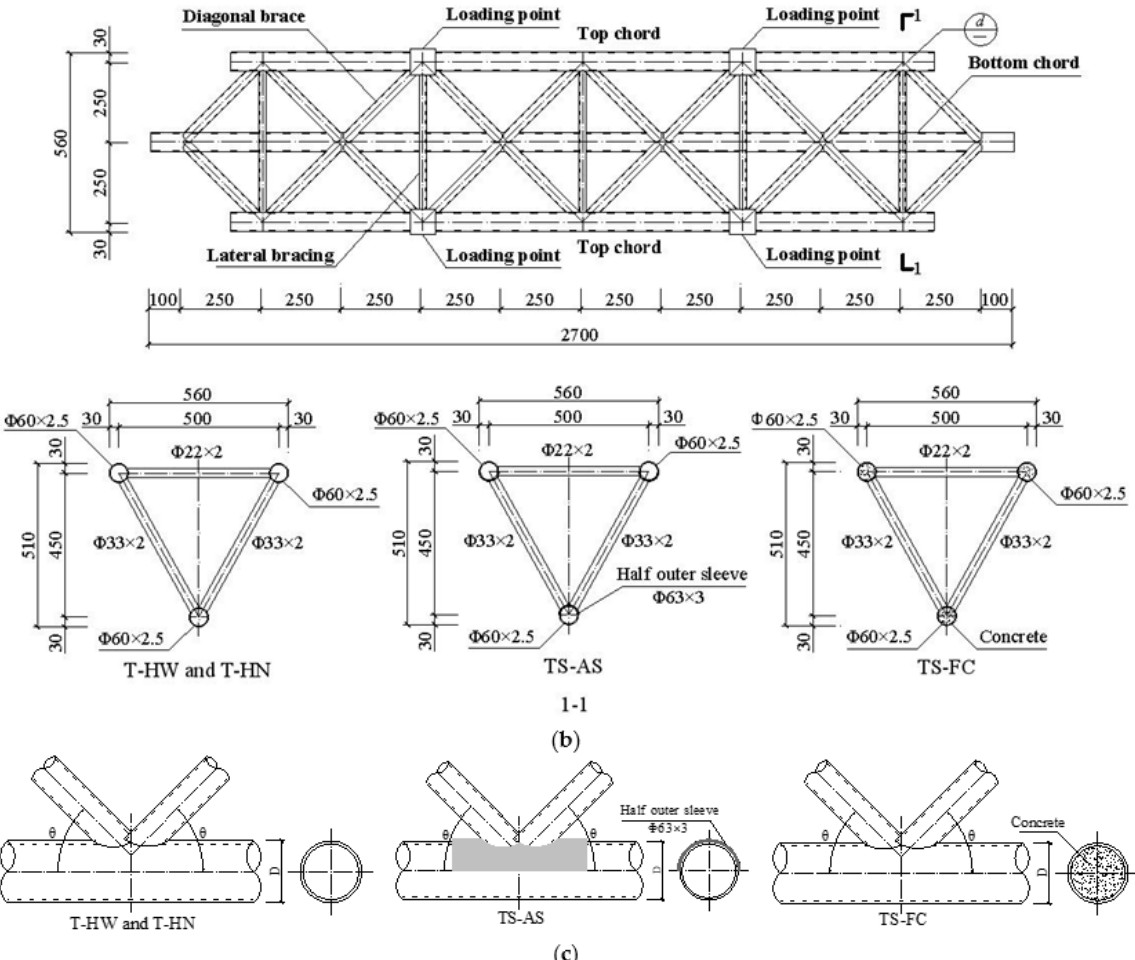

**Figure 1.** Warren circular hollow section (CHS) tubular truss. (**a**) Front view of the tubular truss (mm); (**b**) Top view of the tubular truss (mm); (**c**) Joints detail of the tubular truss.

**Table 1.** Geometric parameters of specimens.

| Specimen ID | Chord $D \times T$ (mm) | Brace $d \times t$ (mm) | Lateral Bracing $d_1 \times t_1$ (mm) | Dimensionless Parameters | | | | |
|---|---|---|---|---|---|---|---|---|
| | | | | $\theta$ | $\beta$ | $\gamma$ | $\tau$ | $O_v$ |
| T-HW | 60 × 2.5 | 33 × 2 | 22 × 2 | 60° | 0.58 | 12 | 0.8 | 39.3% |
| T-HN | 60 × 2.5 | 33 × 2 | 22 × 2 | 60° | 0.58 | 12 | 0.8 | 39.3% |
| TS-AS | 60 × 2.5 | 33 × 2 | 22 × 2 | 60° | 0.58 | 12 | 0.8 | 39.3% |
| TS-FC | 60 × 2.5 | 33 × 2 | 22 × 2 | 60° | 0.58 | 12 | 0.8 | 39.3% |

*2.2. Material Properties*

All specimens were fabricated with Chinese Standard Q235 steel (nominal yield stress $f_y$ = 235 MPa). Tensile coupon tests were conducted according to the test procedures given in the Chinese Standard of Metallic Materials (GB/T 228-2002) [62] to determine the mechanical properties of carbon steel CHS tubes. The material properties obtained from the tensile coupon tests are the elastic modulus ($E$), tensile yield stress ($f_y$), ultimate tensile stress ($f_u$), and elongation. The concrete-filled Warren CHS tubular trusses were fabricated by filling the concrete with nominal cube strength of 25 MPa in the compression top chord members and the intension bottom chord members. The material properties of concrete cubes with the nominal length of 150 mm were prepared and tested based on the recommendations of the Chinese Standard on Ordinary Concrete (GB/T 50081-2002) [63]. The measured concrete cube strength ($f_{cu}$) is 18.3 MPa and the measured elastic modulus ($E_C$) is 27.7 GPa. The obtained material properties data are listed in Table 2.

**Table 2.** Material properties.

| Steel Tube | Elastic Modulus $E$ (N/mm²) | Tensile Yield Stress $f_y$ (MPa) | Ultimate Tensile Stress $f_u$ (MPa) | $f_y/f_u$ | Elongation $\delta$ (%) |
|---|---|---|---|---|---|
| Φ60 × 2.5 | 2.071 × 105 | 268.57 | 430.14 | 0.62 | 17.87 |
| Φ33 × 2 | 1.957 × 105 | 306.21 | 446.04 | 0.69 | 19.29 |
| Φ22 × 2 | 2.123 × 105 | 324.36 | 462.35 | 0.70 | 15.62 |
| concrete type | elastic modulus $E_c$ (N/mm²) | | concrete cube strength (MPa) | | |
| Self-compacting | 2.77 × 104 | | 18.3 | | |

## 2.3. Test Program

A schematic sketch of a Warren CHS tubular truss under total vertical loading is shown in Figure 2. The test setup includes the reaction frame and supports, hydraulic jack (JSYZ-200, Xi'an Oriental Metal Structure, Xi'an, China), and load measuring system (DH3861N, DONGHUA, Jiangsu, China). The test setup was firmly connected to the strong floor. The specimens were simply supported at their extremities to allow both free in-plane rotation and in-plane longitudinal displacement. Compression force was applied at steel blocks welded at one-fourth and three-fourths of the top chord members with 1000 mm in distance as the loading points using two layers of spreader beams. A photo and a schematic of the test setup are shown in Figure 3.

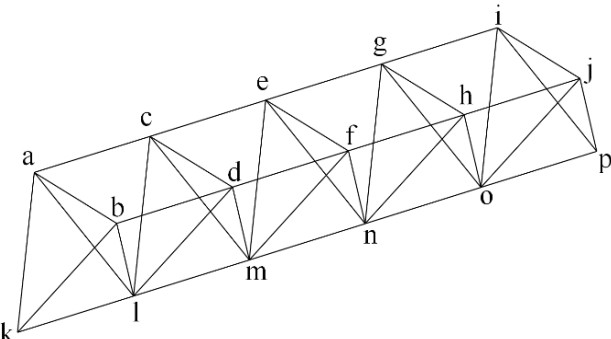

**Figure 2.** Layout of the Warren CHS tubular truss.

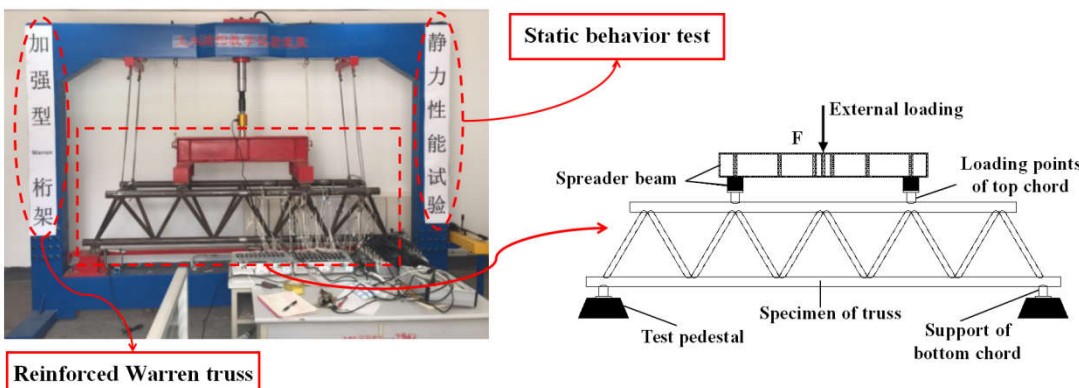

**Figure 3.** Test setup.

An initial load was applied to the specimens to eliminate any possible gaps between the specimens and the loading machine. All specimens were subjected to incremental monotonic static loading, which was equally divided into different load levels. During the tests, the applied loads in each load level were gradually increased to 10 kN when the materials were within the elastic range, while the applied loads in each load level were gradually increased to 2 kN when material plasticity was reached. The readings of the data acquisition system in each load level were recorded with a pause at the

applied loading level for 2 min. The applied loads were then increased to the next loading level and held in place for another 2 min and the readings of the data acquisition system were taken again. This test procedure was repeated until failure of the specimens.

Five displacement transducers D1–D5 (D050, Jing Ming Technology, Yangzhou, China) were used to record the deflections of each specimen during the test. The midspan deflection of the bottom chord members was monitored by displacement transducer D1, and the vertical deflections of the end tubular joints at the bottom chord members were monitored by displacement transducers D2 and D3. In addition, the out-of-plane distortions of specimens were monitored by displacement transducers D4 and D5, which were arranged on the both sides of the midspan of the top chord members. The arrangement of the displacement transducers is shown in Figure 4a.

Two single element strain gauges were attached at the middle length of the brace and chord members at half circumference interval for right half part of the specimens by taking advantage of the symmetry in geometry, loading application, and boundary conditions to obtain the strain intensity of each specimen under different load levels. The arrangement of single element strain gauges is shown in Figure 4b–d for the front view, top view, and bottom view, respectively. The strain along the welded connection of chord and brace members were measured by four multi-axial strain gauges (strain gauge rosettes) attached around the weld toe of connection joints. The arrangement of multi-axial strain gauges is shown in Figure 4e–g for the joints of top chord, midspan joints of bottom chord, and end joints of bottom chord, respectively.

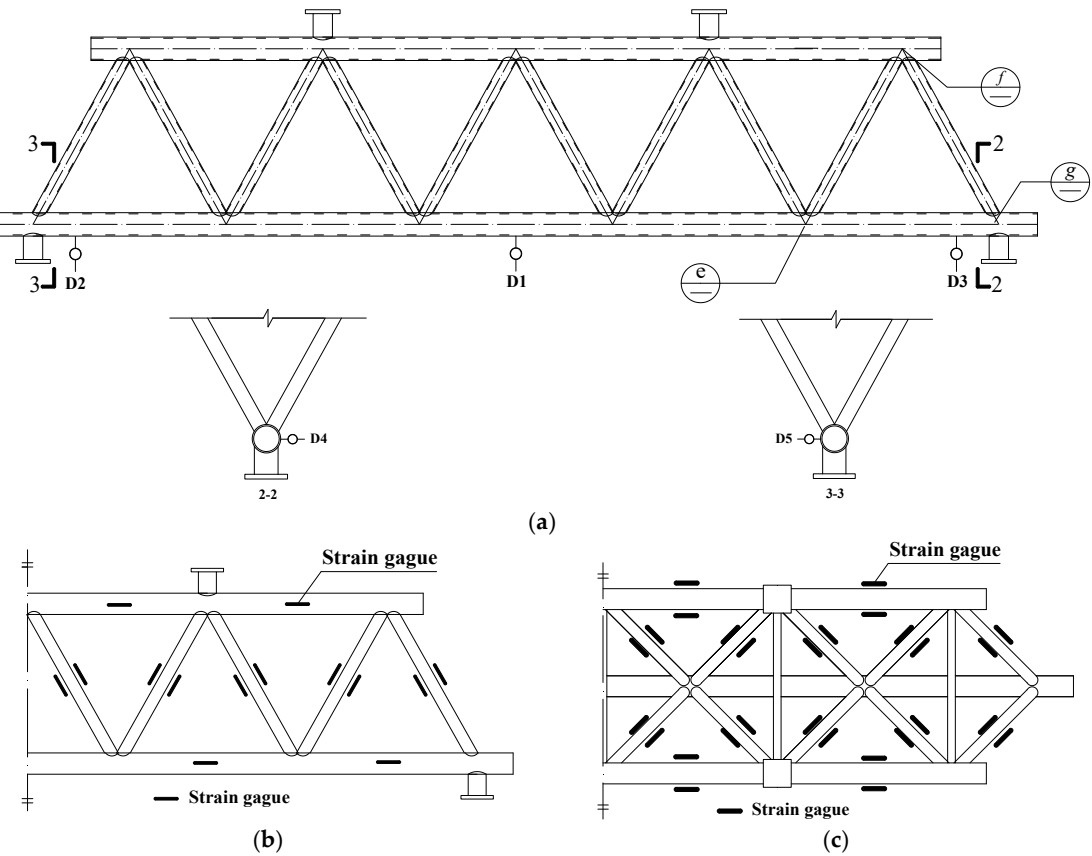

**Figure 4.** *Cont.*

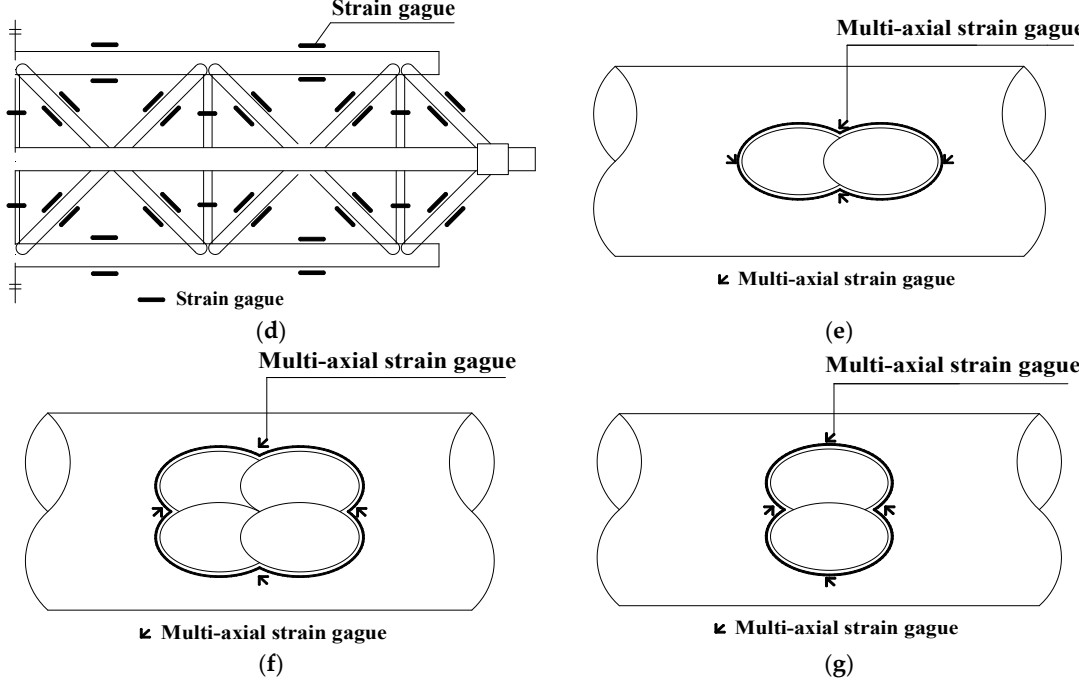

**Figure 4.** Arrangement of displacement transducers and strain gauges. (**a**) Arrangement of the displacement transducers; (**b**) Front view of single element strain gauges; (**c**) Top view of single element strain gauges; (**d**) Bottom view of single element strain gauges; (**e**) Joints of top chord; (**f**) Midspan joints of bottom chord; (**g**) End joints of bottom chord.

## 3. Test Results

### 3.1. Failure Mode

For the layout of all the specimens, as shown in Figure 2, the typical failure modes for the three empty tubular trusses—T-HW, T-HN, and TS-AS—are basically the same, and the bottom chord has different degrees of bending along the span of the truss. For specimen T-HW, distinct concave deformation occurred around the tubular joints 'k' and 'p' at the bottom chord member, and the bending deformation occurred at the bottom chord member. The failure of this specimen resulted from the surface plasticity of the bottom chord member, and the bending of the bottom chord member, as shown in Figure 5. The failure of specimen T-HN was similar to specimen T-HW, while it had a greater degree of the bending deformation of the bottom chord member and surface plastic deformation around the tubular joints 'k' and 'p' at the bottom chord member, as shown in Figure 6. For specimen TS-AS, the concave deformation occurred around the tubular joints 'k' and 'p' at the bottom chord member, and the weld fracture occurred around the tubular joint 'n' along the diagonal brace 'fn', while the local buckling of diagonal brace 'bk' around the tubular joint 'k' was observed. In addition, the bottom chord bent along the truss span. It should be noted that the concave deformation around tubular joints 'k' and 'p' of TS-AS were less than the concave deformation of specimen T-HW and T-HN, while its concave deformation areas of tubular joints 'k' and 'p' were more extensive than that of the T-HW and T-HN specimens. The failure of this specimen resulted from the local buckling of the diagonal brace member, the surface plasticity around the tubular joints at the bottom chord member, and the bending of the bottom chord, as shown in Figure 7. For specimen TS-FC, when reaching the maximum load the test setup could afford, it did not reach failure state and no obvious deformation occurred at each member and joint with slight overall flexure.

It was shown from the tests that the failure modes of the trusses with hidden welds welded or not are basically the same, which are the bending of the bottom chord and surface plasticity of the bottom chord member. In addition, it is shown from the test that TS-AS failed by the local buckling of a

diagonal brace member and the surface plasticity of the bottom chord member because each joint was reinforced with a half outer sleeve on it. The TS-FC specimen did not fail until the test setup reached the maximum load it could impose, which demonstrated that the reinforcement of filling concrete into the top and bottom members made a greater contribution to reinforce the Warren CHS tubular truss than adding a half outer sleeve to each joint. It was shown from the comparison that the failure modes of T-HW and T-HN are different from the two specimens with reinforcement—TS-AS and TS-FC—and the latter two specimens can improve the mechanical behavior and integral stiffness of trusses.

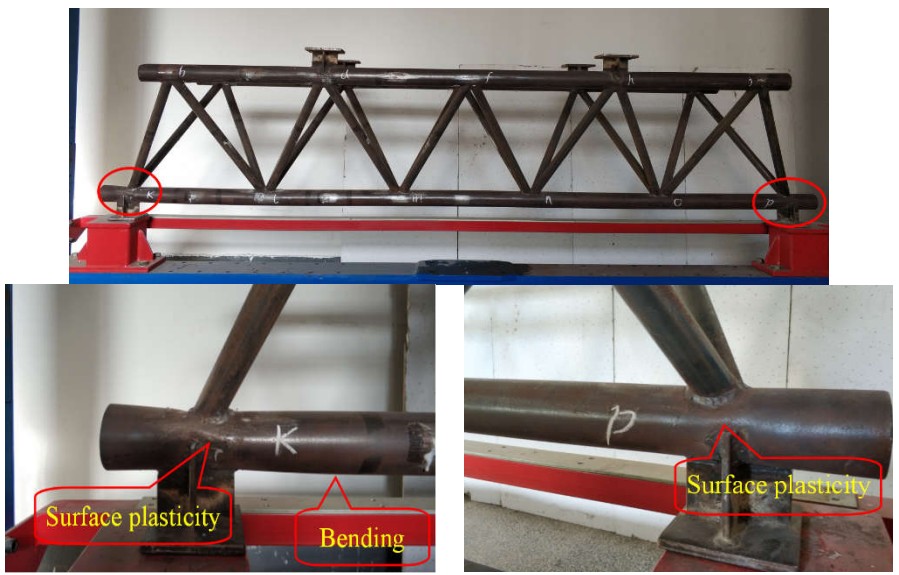

**Figure 5.** Failure modes of specimen T-HW.

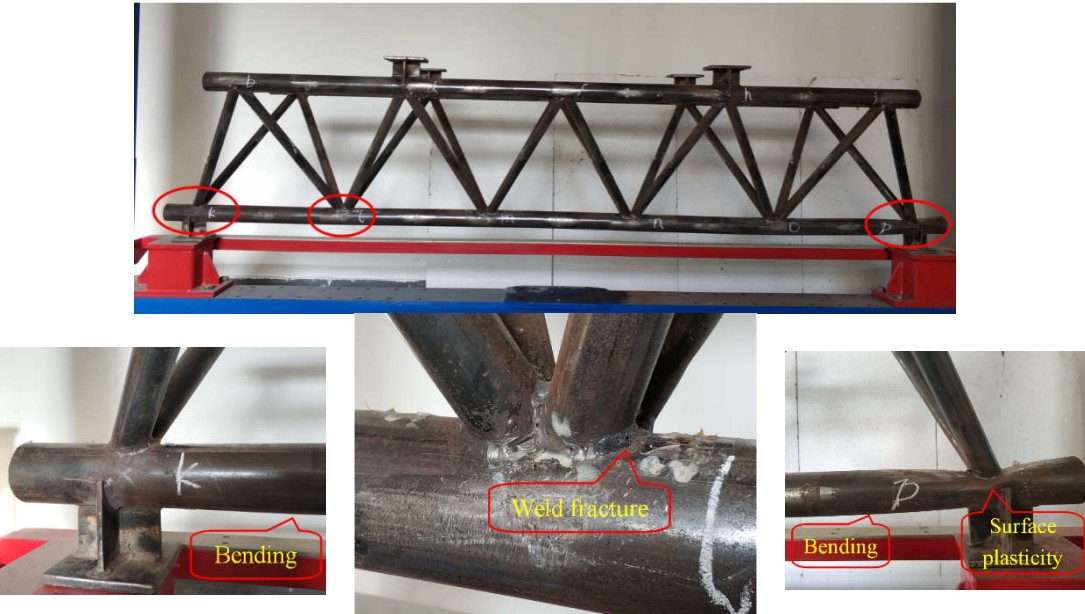

**Figure 6.** Failure modes of specimen T-HN.

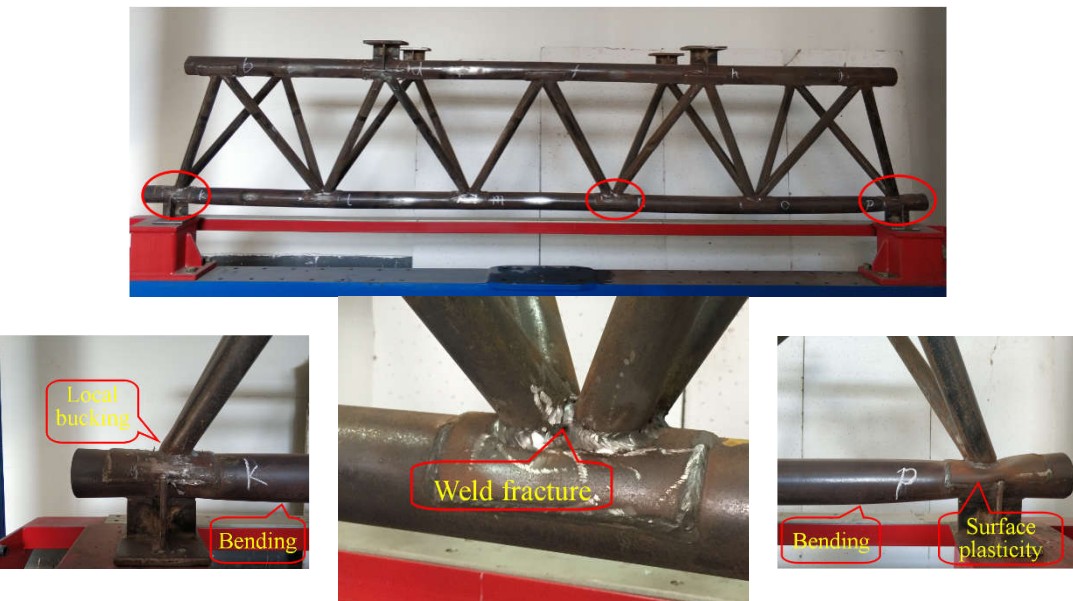

**Figure 7.** Failure modes of specimen TS-AS.

### 3.2. Load Carrying Capacity

The yield loads ($F_y$) which initiate the yielding of truss members and the peak load ($F_p$) of all specimens are summarized in Table 3. It is shown from the comparison that the peak loads ($F_p$) of all specimens were increased roughly by 10% from the yield loads ($F_y$), except for the specimen TS-FC, whose peak load ($F_p$) and yield load ($F_y$) were not obtained because of the limit of test setup, which shows that yield load ($F_y$) and peak load ($F_p$) of specimen TS-FC are the largest among these four specimens. On the other hand, it can be seen from Table 3 that the yield loads ($F_y$) and peak loads ($F_p$) of specimens T-HN, T-HW, TS-AS, and TS-FC are in ascending order. Compared to the load carrying capacity of specimen T-HW, the load carrying capacity of specimen T-HN was decreased by 18%, and the load carrying capacity of specimen TS-AS increased by 36%, whereas the load carrying capacity of specimen TS-FC increased by more than 60%. It is illustrated from the comparison that being hidden weld welded or not influences the load carrying capacity, and the reinforcing method of filling concrete into the top and bottom chord members can improve the load carrying capacity of Warren CHS tubular trusses better than adding a half outer sleeve on each joint, which means the reinforcing method of filling concrete into the top and bottom chord members is much more efficient in practice.

**Table 3.** Load carrying capacity of specimens.

| Specimen | Yield Load $F_y$ (kN) | Peak Load $F_p$ (kN) | Comparison $F_p/F_y$ |
|---|---|---|---|
| T-HW | 100 | 110 | 1.1 |
| T-HN | 80 | 90 | 1.13 |
| TS-AS | 140 | 150 | 1.07 |
| TS-FC | —- | —- | —- |

### 3.3. Overall Deflection

Figure 8 shows the total vertical loading–deflection curves of all specimens. In Figure 8, the horizontal axis represents the vertical deflection ($w$) of the bottom chord member obtained from displacement transducer D1 and D3, and the vertical axis represents the total vertical loading ($P$) applied to the loading points at one-fourth and three-fourths of the top chord members with 1000 mm in distance.

It is shown from the comparison that the midspan deflection of all specimens is greater than that of the joint 'p', however, the changing trends of the initial stiffness and the vertical deflection of all specimens are quite similar. The initial slopes in the elastic stage of the test curves of the T-HN, T-HW, TS-AS, and TS-FC specimens are in ascending order, which means their integral rigidities are in ascending order too. In other words, the reinforcing method of adding a half outer sleeve on each joint and filling concrete into the chord members can effectively improve stiffness of the truss. With the gradual increase of total vertical load, the deformation of the truss develops from the initial overall deflection to local deformation at the joints. The deflections of the TS-AS, T-HN, and T-HW specimens are in descending order, meaning that specimen TS-AS has the best deformability, while specimen T-HN has better deformability than specimen T-HW. Moreover, specimen TS-FC did not fail at the end of the test, and the slope of specimen TS-FC in the elastic stage is larger than that of specimen TS-AS, which means that the reinforcing method of filling concrete into the top and bottom chord members could constrain the deformation at joints more effectively.

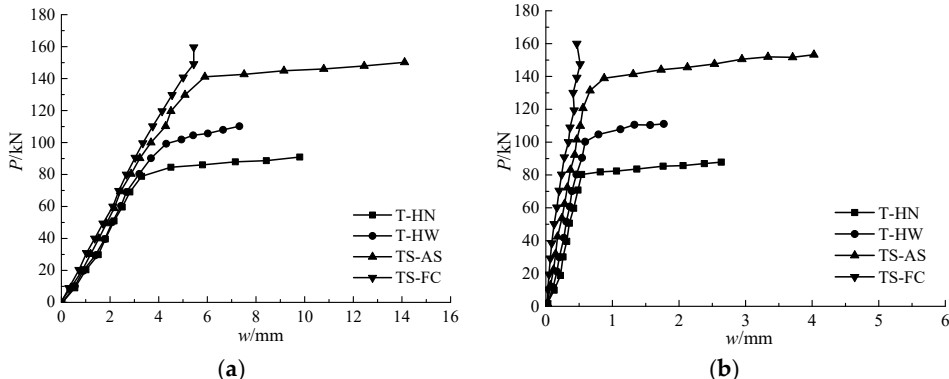

**Figure 8.** Total vertical loading–deflection curves of specimens. (**a**) Midspan (transducer D1); (**b**) End joint 'p' (transducer D3).

### 3.4. Strain Intensity

The total vertical loading versus axial strain curves of all specimens are plotted in Figure 9, in which the horizontal axis represents the axial strain ($\varepsilon$) of joints, brace, and chord members, where positive indicates tensile strain and negative indicates compressive strain, and the vertical axis represents the total vertical loading ($P$) applied to the loading points at one-fourth and three-fourths of the top chord members with 1000 mm in distance. It is illustrated from the comparison that the axial strains of the top chord members in compression are much smaller than the yield strain ($1297 \times 10^{-6}$) of the steel materials, whereas the axial strains of the bottom chord members in tension are smaller than the yield strain of the steel materials, and the axial strains of the diagonal brace members are close to the yield strain ($1565 \times 10^{-6}$) of the steel materials, as shown in Figures 9 and 10 for top chord members, bottom chord member, and diagonal brace members, respectively.

For the top chord members in compression, as shown in Figure 9a–d, the initial slopes of the test curves of specimens T-HW, T-HN, and TS-AS are almost identical, which are smaller to specimen TS-FC. Furthermore, the axial strains of specimens T-HW, T-HN, and TS-AS under the same load level are almost identical, which are larger than that of the specimen TS-FC. For the bottom chord member in tension, as shown in Figure 9e, the test curves of these four specimens are nearly identical. It is shown from the comparison that the filled concrete makes a greater contribution to the top chord members in compression than to the bottom chord member in tension. For the diagonal brace members, as shown in Figure 10, the axial strains of all specimens under different load levels are nearly identical. In addition, the strains at crown positions on the joint 'p' of the specimens T-HW and T-HN vary greatly, which means there is a case of stress concentration, whereas the strains at the same position of specimens TS-AS and TS-FC vary uniformly, as shown in Figure 11. It is shown from the comparison

that the reinforcing methods of adding a half outer sleeve on each joint and filling concrete into the top and bottom chord members could help to improve the force condition of joints. On the other hand, the strains at saddle positions on joint 'n' of the four specimens vary uniformly.

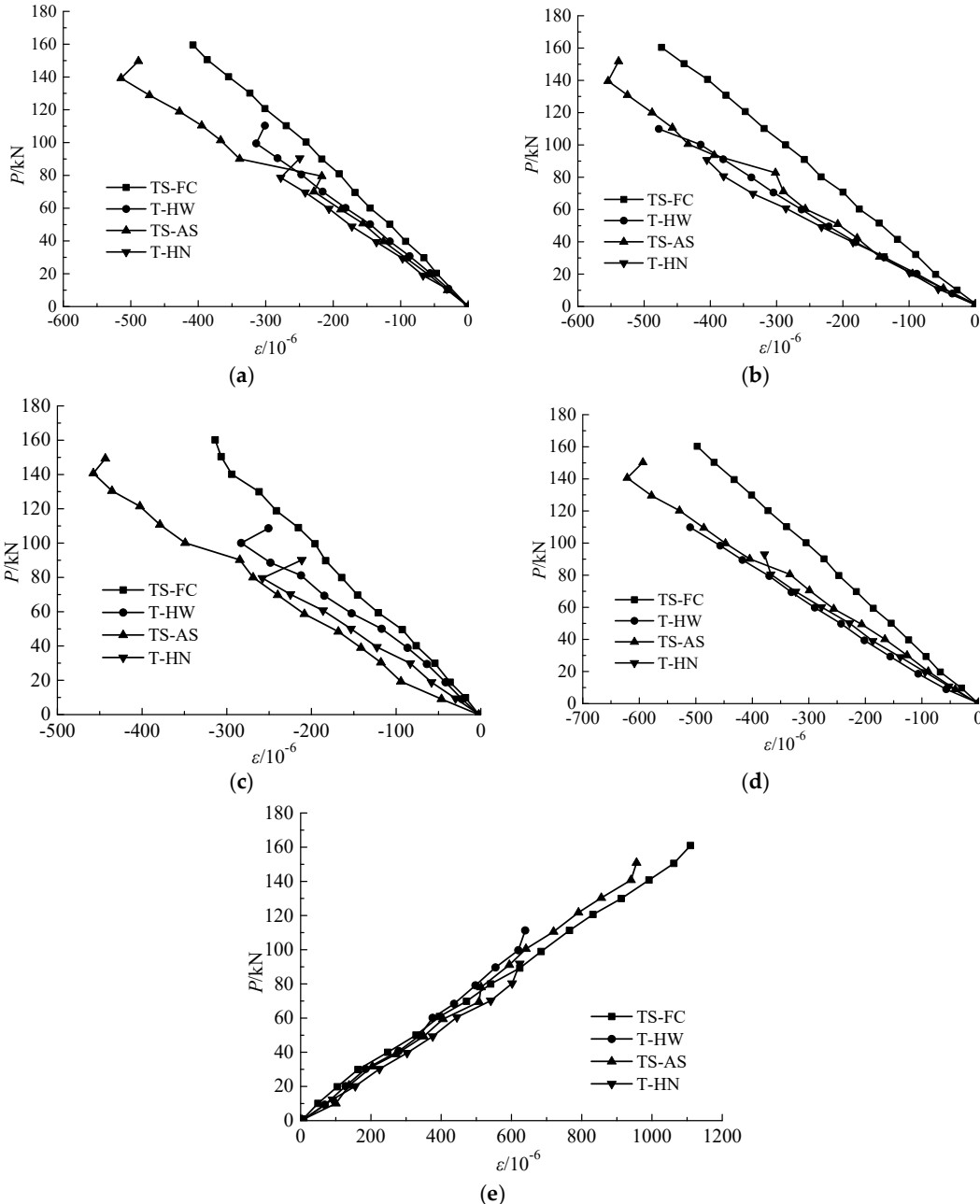

**Figure 9.** Total vertical loading–axial strain curves of chord members. (**a**) Top chord 'hj'; (**b**) Top chord 'fh'; (**c**) Top chord 'gi'; (**d**) Top chord 'eg'; (**e**) Bottom chord 'no'.

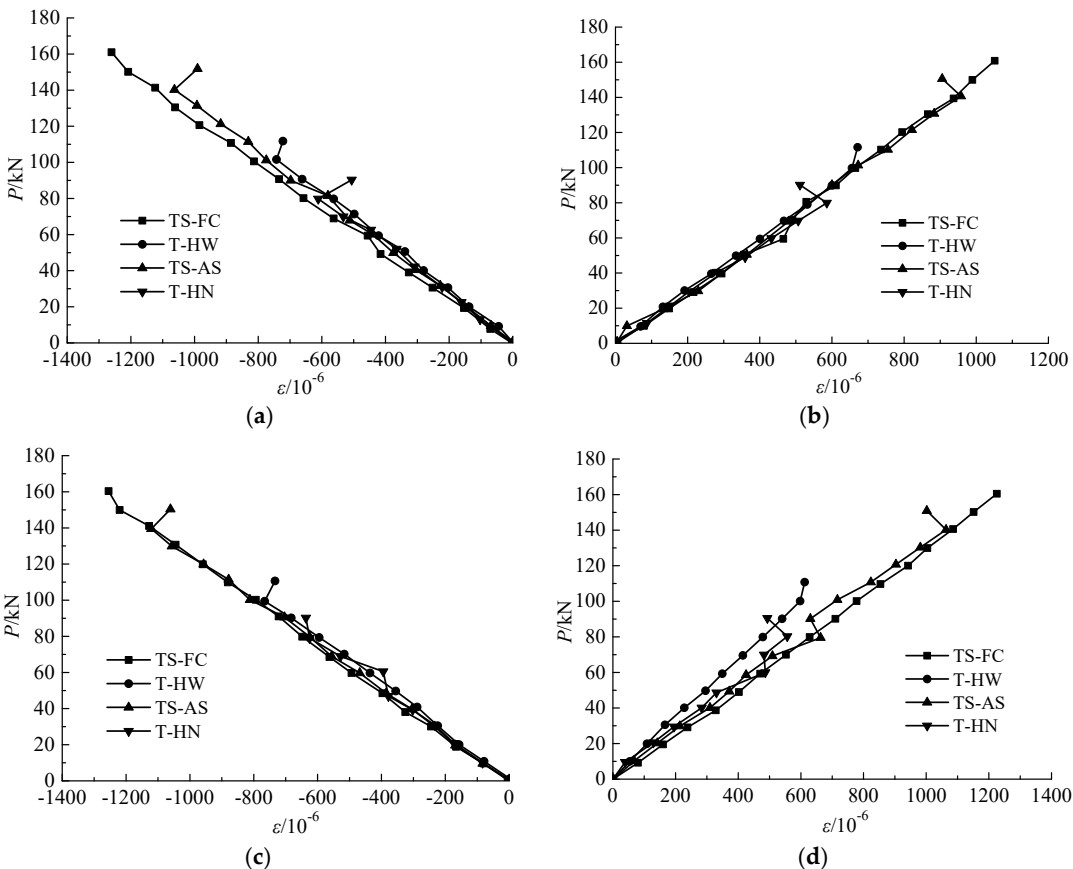

**Figure 10.** Total vertical loading–axial strain curves of diagonal braces. (**a**) Diagonal brace 'jp'; (**b**) Diagonal brace 'jo'; (**c**) Diagonal brace 'ip'; (**d**) Diagonal brace 'io'.

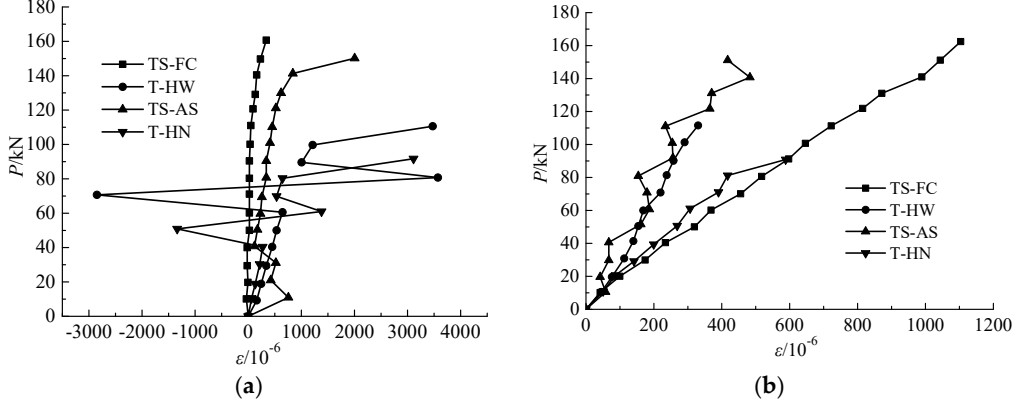

**Figure 11.** Total vertical loading–axial strain curves of the crown and saddle position for joints. (**a**) The crown position of joint 'p'; (**b**) The saddle position of joint 'n'.

## 4. Finite Element Analysis

### 4.1. General

The general purpose finite element program ANSYS [64] (ANSYS 10.0, ANSYS-Inc., Pittsburgh, PA, USA, 2006) was used for the numerical modelling of Warren CHS tubular trusses. The load–displacement nonlinear analysis was performed by using the full Newton–Raphson iteration method available in the ANSYS library. Both material and geometric nonlinearities have been taken into account in the finite element models. The failure modes, load carrying capacities, and overall deflections were obtained from the numerical analysis. The finite element analysis includes various

important factors, such as modelling of materials, finite element type and mesh size, weld model, contact interaction between the steel tube and concrete infill, contact interaction between brace members and chord members, contact interaction between welds and steel tubes, and loading and boundary conditions.

## 4.2. Material Modelling

The bilinear isotropic kinematic hardening model was used for the material modelling of steel tubes. The bilinear stress–strain curve of the steel tube is shown in Figure 12, which includes an elastic stage and a strengthening stage, and the modulus magnitude of the strengthening stage is 1% of that of the elastic stage, in accordance with Shao et al. [65]. The multilinear isotropic hardening plasticity model was adopted for the material modelling of concrete infill, in which the initial part of the multi-linear stress–strain curve represents the elasticity with the measured elastic modulus ($E_c$) of 27.7 GPa and Poisson's ratio ($\nu$) of 0.2, while the tension and compression stress–strain relationship was obtained from the provisions of the Chinese Code for Design of Concrete Structures (GB50010-2010) [66], as shown in Figure 13.

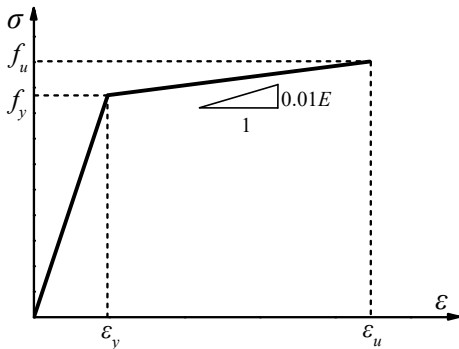

**Figure 12.** Bilinear stress–strain curve of steel tube.

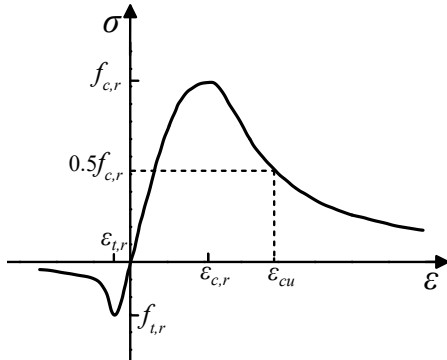

**Figure 13.** Stress–strain model used for confined concrete.

## 4.3. Finite Element Type and Mesh Size

In this study, the Shell181 element available in the ANSYS library was used to model the steel tube of Warren CHS tubular trusses since this element is suitable for analyzing thin to moderately thick shell structures. The Shell181 element is a 4-node element with six degrees of freedom at each node: translations in the x, y, and z axes. In addition, Shell181 is well-suited for linear, larger rotation, and large strain nonlinear applications.

The solid element has been used by many researchers for finite element analysis of concrete-filled tubular trusses. In this study, the Solid65 element available in the ANSYS library was used to model the concrete infill since this element is used for the 3-D modeling of solids without reinforcing bars, and the solid is capable of cracking in tension and crushing in compression. The element Solid65 is

defined by eight nodes with three degrees of freedom at each node: translations in the nodal x, y, and z directions.

In order to guarantee the mesh quality, a sub-zone mesh generation method was used during the finite element modeling. In this method, the entire Warren CHS tubular truss was divided into two different zones. One is the mesh dense zone at every joint, meshed by free meshing. The other is the mesh loose zone between tubular joints, meshed by mapped meshing. The convergence studies were carried out to obtain the optimum finite element mesh size. It was found that the smart size of 10 was appropriate for the mesh dense zone. The ratio of the long side to short side of the mesh element of about 16 could achieve accurate results with minimum computational time. The typical finite element mesh of a Warren CHS tubular truss is shown in Figure 14.

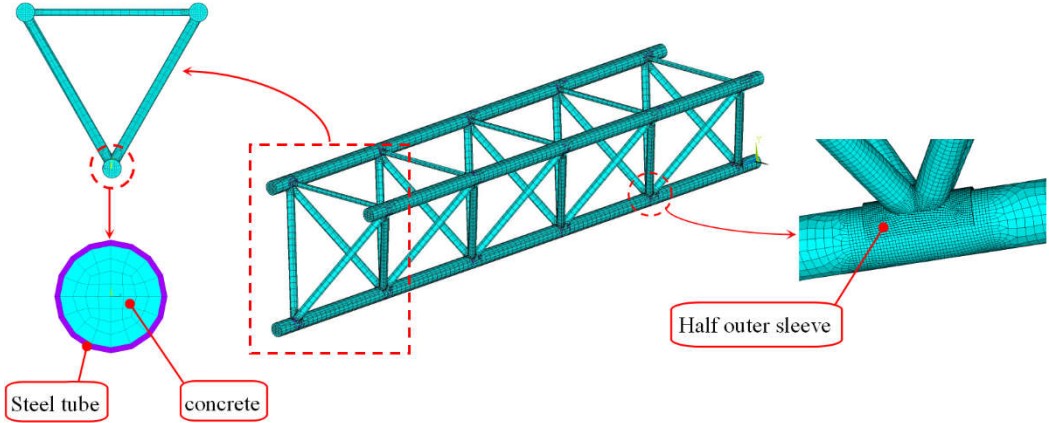

**Figure 14.** Finite element mesh of a Warren CHS tubular truss.

## 4.4. Weld Modeling and Steel Tube–Concrete Infill Interface Model

Weld modeling is seldom taken into consideration in finite element analysis. However, it was found that if the effect of welding is not considered, the numerical simulations of the load carrying capacity and joint rigidity are smaller than those of experimental results [67]. In addition, Corigliano et al. [68,69] pointed out that in the nonlinear finite element analysis of welded joints, the mechanical properties of weld and steel parent metal should be considered differently. In this study, Shell181 was used for weld modeling by adding a circle of Shell181 around the intersection of brace and chord members, and the previous test results of weld metal under monotonic loading were used for the material properties of welds (see reference [35] for details). It should be noted that the contact element was used for simulating the case of hidden weld unwelded. The weld modeling is shown in Figure 15.

The contact interaction between the steel tube and concrete infill was simulated by setting contact pairs with the contact elements. The steel tube was defined to be the target surface and the outer surface of concrete infill was defined to be the contact surface. In addition, the friction coefficient ($\mu$) should be defined to characterize the friction behaviors of the contact interaction between the steel tube and concrete infill. The friction coefficient ($\mu$) was suggested to be from 0.2 to 0.6, and 0.4 was selected in this study.

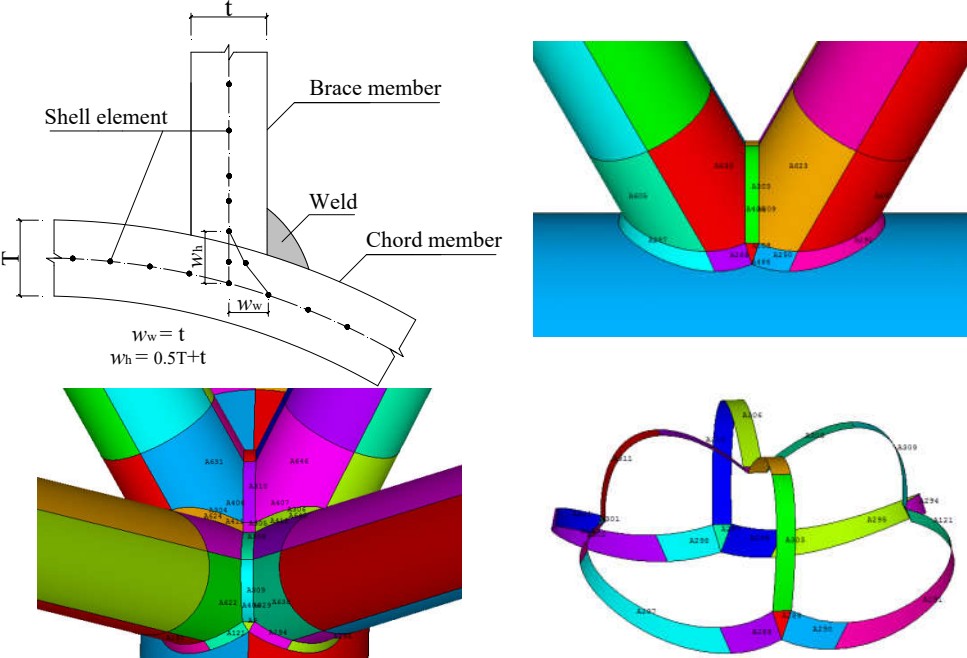

**Figure 15.** Weld modelling.

### 4.5. Loading and Boundary Conditions

In order to make the boundary conditions of finite element model the same as those of the experimental specimens, the end supports at the bottom members and loading points were restrained against all degrees of freedom, except for the displacement at the end supports in the direction of the axis of the bottom chord member and at the loading points in the direction of the applied load. The joints other than the loading points and end supports were free to translate and rotate in any directions.

### 4.6. Verification of Finite Element Models

The midspan vertical deflections under different load levels obtained from the finite element analysis (FEA) of each specimen were compared with the corresponding test results, as shown in Figure 16a–d for the specimens T-HW, T-HN, TS-AS, and TS-FC, respectively. It is shown in Figure 16 that the initial stiffness of all specimens and the change trends in the load–midspan vertical deflection curves showed good agreement. Furthermore, the load carrying capacity of the test for specimen T-HW, T-HN, and TS-AS are 110 kN, 90 kN, and 150 kN, respectively, whereas the load carrying capacity of the FEA for the corresponding specimens are 116 kN, 94 kN, and 158 kN, respectively. For specimen TS-FC, since it did not reach failure, the complete load–midspan vertical deflection curve was not obtained and it was basically in the elastic stage. The load–midspan vertical deflection curve is basically consistent with the corresponding FEA result in the elastic stage, as shown in Figure 16d. Overall, the load carrying capacities of the FEA are slightly higher than those of the experimental test, and the highest difference value between the test and FEA is 5.4%, which meets the requirements of practical engineering. Furthermore, the axial strains of all specimens from the test and FEA were compared in Figure 17, in which the vertical axis represents the axial strain ($\varepsilon$) of chord and brace members at the end of the experimental loading. It can be seen from Figure 17 that the axial strains of the FEA are slightly smaller than the corresponding test, but the difference values between them are relatively small (the largest difference value is 6.1%). It is shown from the comparison that the FEA results of specimens agreed well with the corresponding test results.

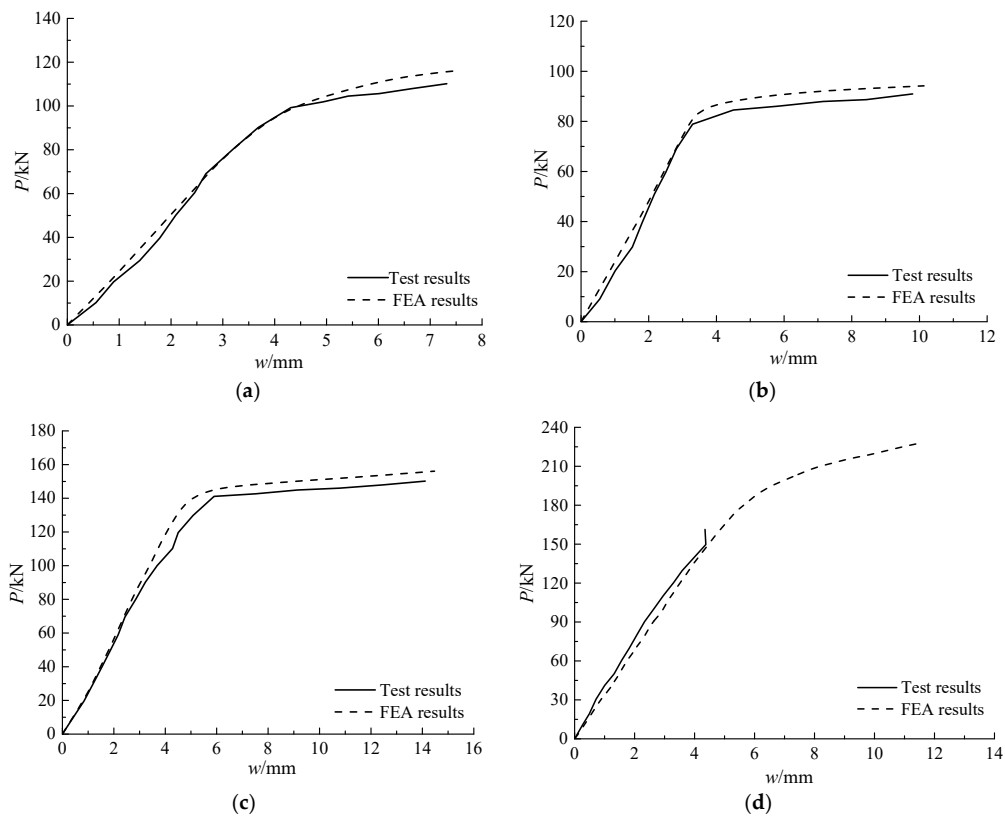

**Figure 16.** Comparison of experimental and finite element analysis vertical load–midspan vertical deflection curves. (**a**) T-HW; (**b**) T-HN; (**c**) TS-AS; (**d**) TS-FC.

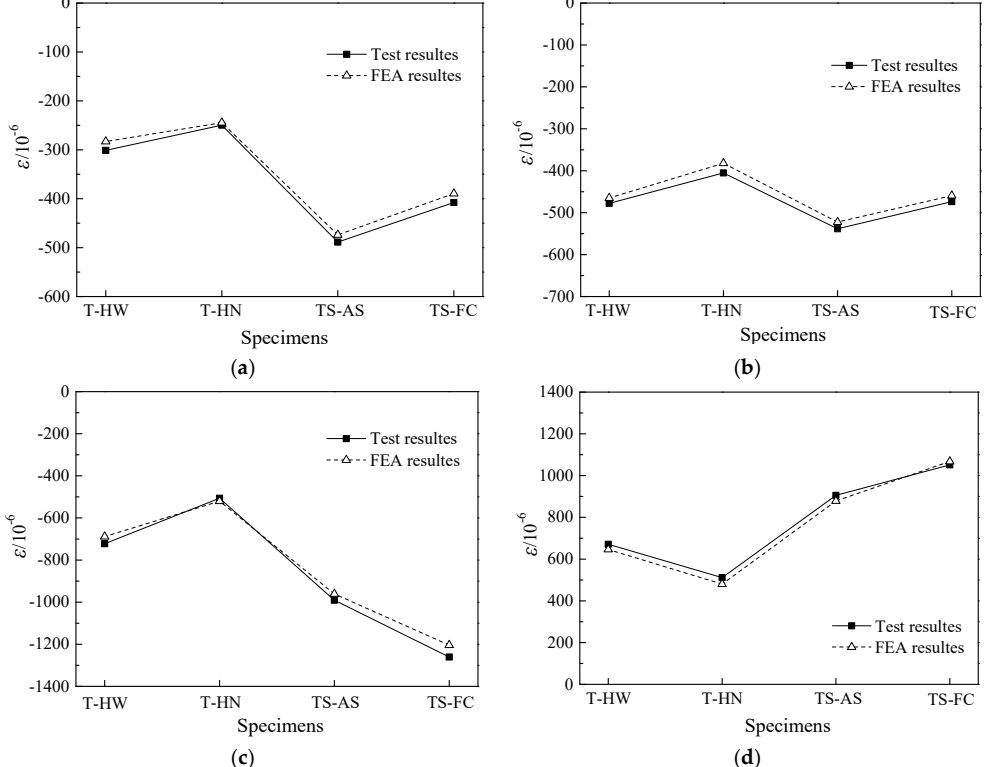

**Figure 17.** Comparison of experimental and finite element analysis axial strains. (**a**) Top chord 'hj'; (**b**)Top chord 'fh'; (**c**) Diagonal brace 'jp'; (**d**) Diagonal brace 'jo'.

On the other hand, the failure modes obtained from the finite element analysis of specimen T-HW were compared with the corresponding test results, as shown in Figures 18 and 19. It is shown from the comparison that good agreement between the experimental and finite element analysis results was achieved, and the main cause of the experiment and FEA failures were the surface plasticity of the bottom chord member in addition to bending deformation along the truss span. Therefore, the developed finite element models were verified to be accurate and reliable. In addition, it is observed from Figures 18 and 19 that the stress distribution law for specimen T-HW at the joint 'k' and joint 'p' are basically the same. The areas of chord surface at the brace–chord intersection became plastic, because the maximum stresses shown in Figures 18 and 19 are 369 MPa and 358 MPa, respectively, which exceed the yield tress of 268 MPa. However, the other areas of chord members do not yield, and are in the elastic state.

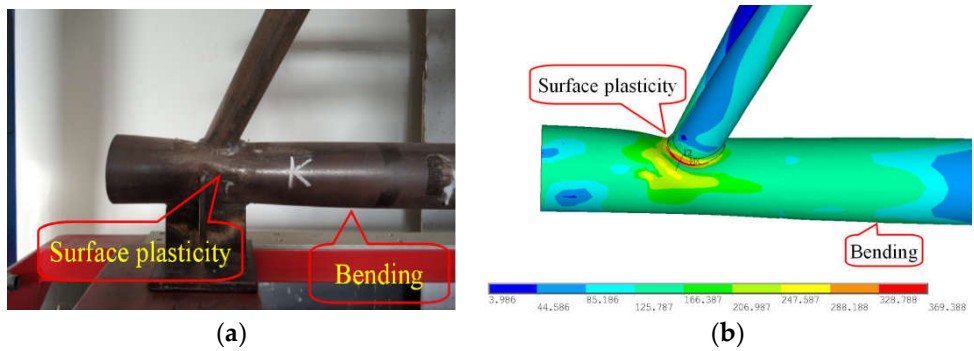

| (**a**) | (**b**) |

**Figure 18.** Comparison of experimental and FEA failure mode for specimen T-HW at the joint 'k'. (**a**) Experimental; (**b**) FEA analysis.

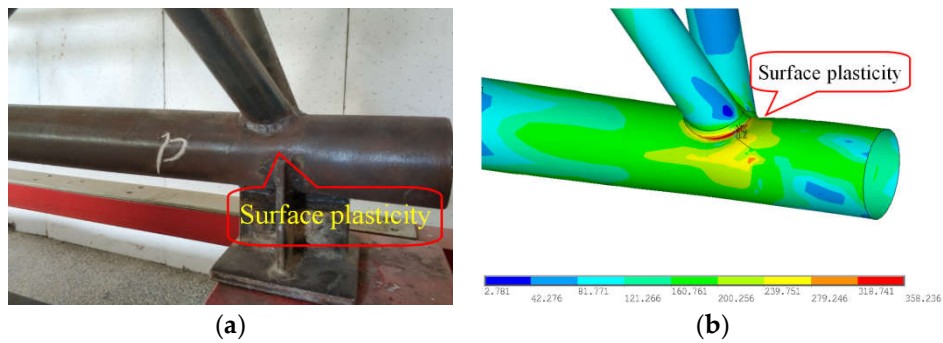

| (**a**) | (**b**) |

**Figure 19.** Comparison of experimental and FEA failure mode for specimen T-HW at the joint 'p'. (**a**) Experimental; (**b**) FEA analysis

## 5. Conclusions and Future Work

An experimental investigation and corresponding numerical investigations were performed in this study on four different Warren CHS tubular trusses, including a truss with its hidden welds welded (T-HW), a truss with its hidden welds unwelded (T-HN), a truss with its hidden welds unwelded but reinforced by adding a half outer sleeve on each joint (TS-AS), and a truss with its hidden welds unwelded but reinforced by filling concrete into the top and bottom chord members (TS-FC). Some conclusions can be drawn from the experimental and numerical investigations as follows:

(1)    The typical failure modes for three empty tubular trusses—T-HW, T-HN, and TS-AS—are basically the same, which resulted from the surface plasticity of the bottom chord member, the weld fracture around tubular joints at the bottom chord member, and bending deformation occurring at the bottom chord. It should be noted that the concave deformation at the end joints of TS-AS were slighter than the concave deformation of T-HW and T-HN, while its concave deformation areas at the end joints were more extensive than for T-HW and T-HN.

(2)    The TS-FC specimen did not fail until reaching the maximum load the test setup could produce, and there was no visible change in appearance. Furthermore, the load carrying capacity and integral rigidity of specimen TS-FC are larger than for specimen TS-AS, which illustrated that the reinforcing method of filling concrete into the top and bottom chord members can improve the mechanical behavior and integral stiffness of trusses better than adding a half outer sleeve on each joint.

(3)    Compared with the load carrying capacity of specimen T-HW, the load carrying capacity of specimen T-HN decreased by 18%, and the load carrying capacity of specimen TS-AS increased by 36%, whereas the load carrying capacity of specimen TS-FC increased by more than 60%. The deflections of the specimens TS-AS, T-HN, and T-HW are in descending order, meaning that specimen TS-AS has the best deformability, while specimen T-HN has better deformability than specimen T-HW.

(4)    The axial strains of the top chord members in compression are much smaller than the yield strain of the steel material, whereas the axial strains of the bottom chord members in tension are smaller than the yield strain of steel materials, and the axial strains of diagonal brace members are close to the yield strain of the steel materials. In addition, the strains at the crown positions on joints of the T-HW and T-HN specimens vary greatly, which means there is a case of stress concentration, whereas the strains at the same position of specimens TS-AS and TS-FC vary uniformly. It is shown that the reinforcing methods of adding a half outer sleeve on each joint and filling concrete into the top and bottom chord members could help to improve the force condition of joints of warren CHS tubular trusses.

(5)    The midspan vertical deflections under different load levels, the failure modes, and the load carrying capacity obtained from the finite element analysis of each specimen were compared with the corresponding test results. Good agreement between the experimental and finite element analysis results was achieved and the finite element models were accurate and reliable to be the basis of parametric analysis.

With the recent development of piezoceramic transducer-based structural health monitoring and damage detection in civil structures [70–74], especially in concrete-filled tubular structures [75–80], as a future work, the authors will explore health monitoring and damage detection of Warren CHS tubular trusses using easily installed piezoceramic patch transducers.

**Author Contributions:** W.Y. incepted the original idea and designed the experiment. N.G. analyzed the data. N.-n.G., J.L., and R.Y. wrote the paper. W.Y. revised and finalized the paper.

**Funding:** This research was supported in part by the National Natural Science Foundation of China (No. 51468054) and in part by the Project of Key Research and Development of Ningxia Province (No. 2016KJHM38), also in part by the Funding Project of First-Class Discipline Construction of Universities in Ningxia (Domestic First-Class Discipline Construction) under Grant NXYLXK2017A03.

**Conflicts of Interest:** The authors declare no conflict of interest.

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
