# Peer review of "Experimental Study on the Static Behavior of Reinforced Warren Circular Hollow Section (CHS) Tubular Trusses"

_applsci, doi:10.3390/app8112237_

Round 1
Reviewer 1 Report
The manuscript is in good quality, however, the following revisions needs to be made:
(1) The english language of the manuscript needs to be improved.
(2) Please explain why you chose the static analysis instead of the dynamic one.
(3) The introduction and literature review needs to be updated with the research of other scientists on truss bridges. You may want to refer to the following publication:
Khademi, F. (2017). Enhancing Load Rating of Railway Truss Bridges through a Hybrid Structural Analysis and Instrumentation Procedure (Doctoral dissertation, Illinois Institute of Technology).
(4) In the theoretical truss design, most of the time the connections are pinned connections. Please explain why you preferred weld connections instead of the pinned ones?
The overall quality of the manuscript is in good condition. My suggestion is minor revision. After these revisions are made, the manuscript has a high chance of getting accepted in the journal.
Author Response
Dear Reviewer:
Thanks very much for your hard work. We have revised the manuscript according to your kind advices and referee’s detailed advice, and the amendments are highlighted in red in the revised manuscript. We sincerely hope this manuscript will be finally acceptable to be published. Thank you very much for all your help and looking forward to hearing from you soon.
Best regards
Sincerely yours
We submit here the revised manuscript as well as a list of changes:
Response to Reviewer 1 Comments
Point 1: The English language of the manuscript needs to be improved.
Response 1: Thanks very much for the suggestions from the referee, which are very helpful for us to improve the manuscript, and our language should be improved. After carefully check, we found many grammar and sentence errors, and have modified the manuscript accordingly. and we hope the revised paper will be more clear and accurate on expressions.
Point 2: Please explain why you chose the static analysis instead of the dynamic one.
Response 2:The mechanical behavior analysis of the truss structure mainly includes static analysis and dynamic analysis. Since the structure is subjected to static loads for a long time during its service, it is necessary for the static analysis of the structure. Furthermore, the results of static analysis can also provide reference for structural dynamic analysis. Therefore, the research in this paper mainly focuses on the static performance of the truss structure. In the future, we will carry out further research on the dynamic behavior of truss structure.
Point 3: The introduction and literature review needs to be updated with the research of other scientists on truss bridges. You may want to refer to the following publication:
Khademi, F. (2017). Enhancing Load Rating of Railway Truss Bridges through a Hybrid Structural Analysis and Instrumentation Procedure (Doctoral dissertation, Illinois Institute of Technology).
Response 3:Wequite agree with the suggestion on updating the research of other scientists on truss bridges. This is very helpful to the further improvement of our manuscript.And some recent references which suggested by reviewers are quoted. These two references are numbered as 9 and 10.
Point 4:In the theoretical truss design, most of the time the connections are pinnedconnections. Please explain why you preferred weld connections instead of the pinned ones?
Response4: With regard to the above problem which is mentioned by reviewer, we can explain from the following two aspects:
1)The connection form of truss structure. The weld connection has the advantages of better tightness, large rigidity, convenient construction and mature technology. Furthermore, with the development of CNC (Computer Numerical Control) cutting equipment, fabrication of directly welded joints becomes easier, and weld connections become the common connection form of truss structures.
2)Thetheoretical design of truss structure. Since the larger slenderness ratio of the member makes the bending moment of the joint smaller, the connections of the truss can be simplified to the pinnedconnections in the theoreticaldesign. On the other hand, the connection between members is welded in practical engineering, while the conservative pinnedconnections are used in the theoretical design, which can make the truss structure have a high safety reserve.
Reviewer 2 Report
The paper represents a very good contribution in the field of experimental and FE simulations of full-scale structures, furthermore the interaction of steel and concrete is taken into account.
The reviewer believes that this paper can be accepted after some minor amendments which will increase the quality of the paper:
1) Material modelling : The authors wrote “The bilinear stress-strain curve of steel tube is shown in Figure 12, which includes an elastic 328 stage and a strengthening stage. In addition, the modulus magnitude of the strengthening stage is 1% of that of the elastic stage.” The authors should tell where this statement came from and eventually cite it as a reference.
2) About the weld modelling: when dealing with elasto-plastic FE simulations, especially in the plastic simulation region ,the weld mechanical properties differ from the steel parent metal, (see : 1- P. Corigliano, V. Crupi, E. Guglielmino, W. Fricke.“FE analysis of cruciform welded joints considering different mechanical properties for base material, heat affected zone and weld metal” Fracture and Structural Integrity. 2- P. Corigliano, V. Crupi, W. Fricke, N. Friedrich, E. Guglielmino. “Experimental and numerical analysis of fillet-welded joints under low-cycle fatigue loading by means of full-field techniques”. Journal of Mechanical Engineering Science, Proceedings of the Institution of Mechanical Engineers Part C). How did the authors take into account this aspect? If they did not modelled different mechanical properties for Weld metal, Heat Affected Zone and Parent Metal they should mention that this is an approximation in the paper.
3) Verification of finite element models: the scale in the FE results if figures 17 and 18 should be added. Furthermore the authors compared only the midspan deflection quantitatively; afigure reporting the comparison in terms of strains, with some of the strain gauges, will give more value to the FE validation.
Author Response
Dear Reviewer:
Thanks very much for your hard work. We have revised the manuscript according to your kind advices and referee’s detailed advice, and the amendments are highlighted in red in the revised manuscript. We sincerely hope this manuscript will be finally acceptable to be published. Thank you very much for all your help and looking forward to hearing from you soon.
Best regards
Sincerely yours
We submit here the revised manuscript as well as a list of changes:
Response to Reviewer 2 Comments
Point 1: Material modelling : The authors wrote “The bilinear stress-strain curve of steel tube is shown in Figure 12, which includes an elastic stage and a strengthening stage. In addition, the modulus magnitude of the strengthening stage is 1% of that of the elastic stage.” The authors should tell where this statement came from and eventually cite it as a reference.
Response1: Thanks for the reminding by the referee, we agree with that the correct citation of reference is a very important link in thesis writing. The relevant reference is cited, and it is numbered as 68.
Point 2:About the weld modelling: when dealing with elasto-plastic FE simulations, especially in the plastic simulation region ,the weld mechanical properties differ from the steel parent metal, (see : 1- P. Corigliano, V. Crupi, E. Guglielmino, W. Fricke.“FE analysis of cruciform welded joints considering different mechanical properties for base material, heat affected zone and weld metal” Fracture and Structural Integrity. 2- P. Corigliano, V. Crupi, W. Fricke, N. Friedrich, E. Guglielmino. “Experimental and numerical analysis of fillet-welded joints under low-cycle fatigue loading by means of full-field techniques”. Journal of Mechanical Engineering Science, Proceedings of the Institution of Mechanical Engineers Part C). How did the authors take into account this aspect? If they did not modelled different mechanical properties for Weld metal, Heat Affected Zone and Parent Metal they should mention that this is an approximation in the paper.
Response2: According your guidance, we quoted some references (70 and 71) which are suggested by you to illustrate that in the nonlinear finite element analysis of welded joints, the weld mechanical properties differ from the steel parent metal. In this paper, the mechanical properties of weld and steel parent metal are considered differently. The previous test results of weld metal under monotonic loading were used for the material properties of weld, and the amendments are highlighted in red.
Point 3: Verification of finite element models: the scale in the FE results if figures 17 and 18 should be added. Furthermore the authors compared only the midspan deflection quantitatively; afigure reporting the comparison in terms of strains, with some of the strain gauges, will give more value to the FE validation.
Response3: We quite agree with the suggestions from reviewer, which are very helpful for us to further improve the manuscript. In the revised manuscript, the scale of the FE results shown inFigures 18 and 19have been added, and Figure 17 have been supplemented to compare the strain values of the test and FE analysis.
Reviewer 3 Report
This document is clearly written, well organized and includes an appropriate state-of-the-art. Results are justified by both experimental and numerical analysis. Furthermore, the topic -static behavior of different reinforced Warren circular hollow section tubular truss- is of interest to the readers. In my opinion, this paper can be accepted for publication in present form.
Author Response
Dear Reviewer:
Thanks very much for your hard work. We have revised the manuscript according to your kind advices and referee’s detailed advice, and the amendments are highlighted in red in the revised manuscript. We sincerely hope this manuscript will be finally acceptable to be published. Thank you very much for all your help and looking forward to hearing from you soon.
Best regards
Sincerely yours